# Association between Obesity and COVID-19 Mortality in Peru: An Ecological Study

**DOI:** 10.3390/tropicalmed6040182

**Published:** 2021-10-07

**Authors:** Max Carlos Ramírez-Soto, Miluska Alarcón-Arroyo, Yajaira Chilcon-Vitor, Yelibeth Chirinos-Pérez, Gabriela Quispe-Vargas, Kelly Solsol-Jacome, Elizabeth Quintana-Zavaleta

**Affiliations:** Facultad de Ciencias de la Salud, Universidad Tecnologica del Peru, 15046 Lima, Peru; U18205082@utp.edu.pe (M.A.-A.); U19102232@utp.edu.pe (Y.C.-V.); U18301301@utp.edu.pe (Y.C.-P.); U18305711@utp.edu.pe (G.Q.-V.); U18302765@utp.edu.pe (K.S.-J.); U19205940@utp.edu.pe (E.Q.-Z.)

**Keywords:** COVID-19, pandemic, overweight, obesity, Peru

## Abstract

There is a gap in the epidemiological data on obesity and COVID-19 mortality in low and middle-income countries worst affected by the COVID-19 pandemic, including Peru. In this ecological study, we explored the association between body mass index (BMI), the prevalence of overweight and obesity, and the COVID-19 mortality rates in 25 Peruvian regions, adjusted for confounding factors (mean age in the region, mean income, gender balance and number of Intensive Care Unit (ICU) beds) using multiple linear regression. We retrieved secondary region-level data on the BMI average and prevalence rates of overweight and obesity in individuals aged ≥ 15 years old, from the Peruvian National Demographics and Health Survey (ENDES 2020). COVID-19 death statistics were obtained from the National System of Deaths (SINADEF) from the Peruvian Ministry of Health and were accurate as of 3 June 2021. COVID-19 mortality rates (per 100,000 habitants) were calculated among those aged ≥ 15 years old. During the study period, a total of 190,046 COVID-19 deaths were registered in individuals aged ≥ 15 years in 25 Peruvian regions. There was association between the BMI (*r* = 0.74; *p* = 0.00001) and obesity (*r* = 0.76; *p* = 0.00001), and the COVID-19 mortality rate. Adjusted for confounding factors, only the prevalence rate of obesity was associated with COVID-19 mortality rate (*β* = 0.585; *p* = 0.033). These findings suggest that as obesity prevalence increases, the COVID-19 mortality rates increase in the Peruvian population ≥ 15 years. These findings can help to elucidate the high COVID-19 mortality rates in Peru.

## 1. Introduction

Overweight and obesity are global public health problems [1]. Their prevalence has increased rapidly during recent decades [2,3], and studies have shown an association between obesity and infectious diseases [4]. During the COVID-19 pandemic, studies in high-income countries have shown that obesity increases the risk for hospitalization and death among patients with COVID-19 [5,6]. Most studies included patients with COVID-19 symptoms admitted to hospital, where obesity itself and the severity of the disease increase the risk of death [7,8]. Avoiding this bias, one recent study showed that excess weight linearly increased the risk of severe COVID-19, leading to admission to hospital and death (body mass index > 28 kg/m^2^) [9]. In other observational studies, obesity prevalence was significantly correlated with both infection and/or COVID-19 mortality [10,11,12,13]. Despite these findings, to date, there is a gap of epidemiological data on obesity and COVID-19 mortality in low- and middle-income countries.

In Latin American, Peru has been one of the worst-affected countries by the COVID-19 pandemic [14,15]. Despite the rapid implementation of control measures, by the end of June 2021, more than two million cases and over 190,000 deaths were confirmed, with a case fatality rate of 9.31% [16]. In addition, a study in the first months of the COVID-19 pandemic found a correlation between the prevalence of obesity and COVID-19 mortality, although its findings are limited [13]. Despite this, it has not been documented how COVID-19 mortality rates vary according to body mass index (BMI) and the prevalence of overweight and obesity. Here, we explored the association between body mass index (BMI), the prevalence of overweight and obesity, and COVID-19 mortality in 25 Peruvian regions, adjusted by for possible confounding factors.

## 2. Materials and Methods

### 2.1. Study Design and Setting

We performed an ecological study following the Strengthening the Reporting of Observational Studies in Epidemiology (STROBE) reporting guidelines [17]. For this study, we retrieved secondary region-level data on the BMI average and prevalence rates of overweight and obesity in individuals aged ≥ 15 years old from the Peruvian National Demographics and Health Survey (ENDES 2020) [18]. ENDES 2020 includes a sample of 32,197 men and women aged 15 years or more, from 25 Peruvian regions (Figure 1), from January to December 2020. COVID-19 deaths were obtained from the National System of Deaths (SINADEF) from the Peruvian Ministry of Health (MINSA), accurate as of 3 June 2021 [19]. The SINADEF database records all deaths that occur in Peru and generates the death certificates and statistical reports. Death records with COVID-19 as the underlying cause of death were included in the study. Data on the mean age in the region, mean income and gender balance were retrieved obtained from the National Institute of Statistics and Informatics (INEI). The number of Intensive Care Unit (ICU) beds was obtained of App. F500.2 from at the Superintendencia Nacional de Salud, Perú (SUSALUD).

### 2.2. Statistical Analysis

COVID-19 mortality rates (per 100,000 habitants) among those aged ≥ 15 years old were calculated by dividing the number of COVID-19 deaths per department by the estimated population of each department. Population counts for calculating mortality rates were obtained from the INEI, Peru [20]. Spearman’s test and linear regression models were used to estimate correlations between the BMI, the prevalence of overweight and obesity, and COVID-19 mortality rates. Multiple regression analysis was also used for possible confounding factors. *p*-values < 0.05 were considered significant. Confounding factors included the mean age in the region (years), mean monthly income (PEN), gender balance and number of ICU beds. Statistical analyses were conducted using StataSE 16.0 Software.

This study was based on public use data that do not include personal information; therefore, it was exempt from institutional review board approval.

## 3. Results

During the study period, a total of 190,046 COVID-19 deaths were registered in individuals aged ≥ 15 years in 25 Peruvian regions. Among the individuals aged ≥ 15 years old, the highest prevalence rates of overweight and obesity were registered in Tacna, Moquegua, and Ica regions. The five regions with the highest COVID-19 mortality rates were Ica (1083.6 per 100,000 habitants), Callao (1071.3 per 100,000 habitants), Lima (979.9 per 100,000 habitants), Moquegua (883.3 per 100,000 habitants), and Lambayeque (811.4 per 100,000 habitants) (Table 1).

There was an association between BMI (*r* = 0.74; *p* = 0.00001) and obesity (*r* = 0.76; *p* = 0.00001) and the COVID-19 mortality rate (per 100,000 habitants) (Figure 2A–C). Adjusted by possible confounding factors (mean age in the region, mean monthly income, gender balance and number of ICU beds), only the prevalence rate of obesity was associated with COVID-19 mortality rate (*β* = 0.585; *p* = 0.033) (Table 2). The model was statistically significant (*F* (5,19) = 8.89, *p* < 0.0002, Adj. R^2^ = 0.60). 

## 4. Discussion

During the COVID-19 pandemic, older adults and people with co-morbidities, including patients with obesity, have experienced the highest risk of COVID-19 death [6,7,8,21]. A previous cohort study reported that, compared with patients with a BMI of 18.5–24 kg/m^2^, patients with BMI ≥ 40 kg/m^2^ had a higher risk of COVID-19 death [7]. Recently, another study reported that the risk of COVID-19 death was more strongly associated with people with a BMI of more than 28 kg/m^2^ [9]. Our findings, despite being correlational, support this observation, because in the Peruvian regions where the BMI was higher (i.e., Tacna, Moquegua and Ica), we found higher COVID-19 mortality rates. However, adjusted by possible confounding factors, there was not association between the BMI and COVID-19 mortality rates. To date, there is little evidence on the mechanisms attributable to BMI effects on COVID-19 mortality. A possible explanation is that excess weight can cause the metabolic impairment of organ functioning [22]. Other possible explanations for BMI increasing COVID-19 mortality could be associated with the severity of COVID-19, male sex, increasing age, and other factors that were not investigated in this study.

In our study, there was correlation between the prevalence of obesity and COVID-19 mortality, i.e., as the obesity prevalence increased, the COVID-19 mortality rates increased in the population aged ≥ 15 years. These findings are consistent with correlational studies which found that, as the obesity prevalence increased, the COVID-19 deaths increased [10,11,12,13]. Cohort studies and meta-analyses on the effect of excess weight on COVID-19 clinical outcomes also reported that obesity was independently associated with the severity of COVID-19 and the risk of death increased [7,8]. The increase in COVID-19 mortality in patients with obesity could be explained by associations with hypertension, diabetes, or respiratory distress syndrome [7,9]; however, to date, the mechanisms explaining the association between obesity and COVID-19 mortality remain limited. In Peru, obesity and high COVID-19 mortality also could be explained by other external factors, such as infrastructure, overload of the health system, medicines, and available intensive care unit beds [14,15].

As with any observational study, the limitations of our study include its ecological design, because we used several different information sources (ENDES 2020, SINADEF, INEI, and SUSALUD); therefore, our findings could have resulted in a possible bias. In our findings, there could also have been an overestimation due to unmeasured covariates, such as the population density, and delayed COVID-19 death registration. On the other hand, the ecological design (group level variables) made it difficult to determine causality between the obesity and the COVID-19 mortality, i.e., we cannot make causal inference with the average characteristics of the group about individual risk. Therefore, our findings should be interpreted at the population level, not the individual-level. Finally, the ecological design is not able to account for changes that impact transmission dynamics, such as the appearance of new variants of concern or the introduction of vaccination. Despite these limitations, the main strengths of our study were the large number of deaths included (*n* = 190,046, as of June 2021) for estimating the COVID-19 mortality and the multiple regression analyses for possible confounding factors (mean age in the region, mean monthly income, gender balance and number of ICU beds), compared with a previous study that included a total of 51,789 deaths (as of July 2020) [13].

## 5. Conclusions

Our findings suggest that, as the obesity prevalence increases, the COVID-19 mortality rates increase in Peruvian populations aged ≥ 15 years. Interventions for obesity improve weight loss; however, we cannot assure that these interventions might reduce COVID-19 mortality. In the long term, there is a need to strive towards achieving healthy weights in the Peruvian population and to decrease the risk of death from other infections in future.

## Figures and Tables

**Figure 1 tropicalmed-06-00182-f001:**
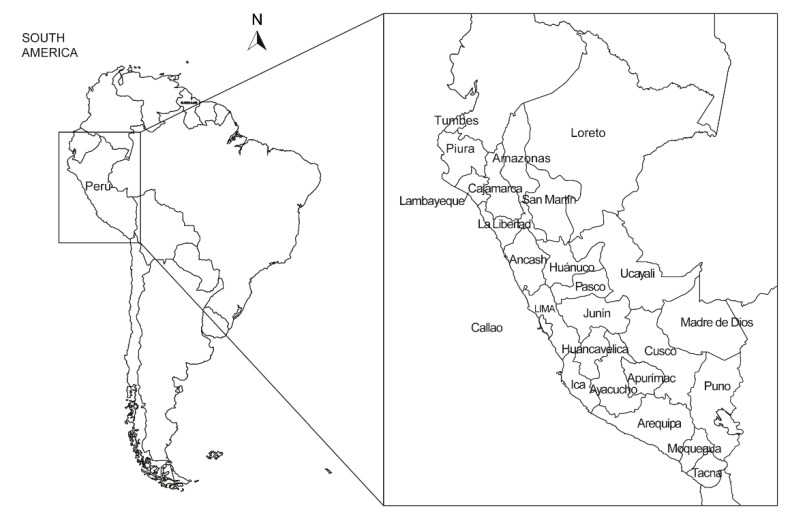
Location of Peru within South American.

**Figure 2 tropicalmed-06-00182-f002:**
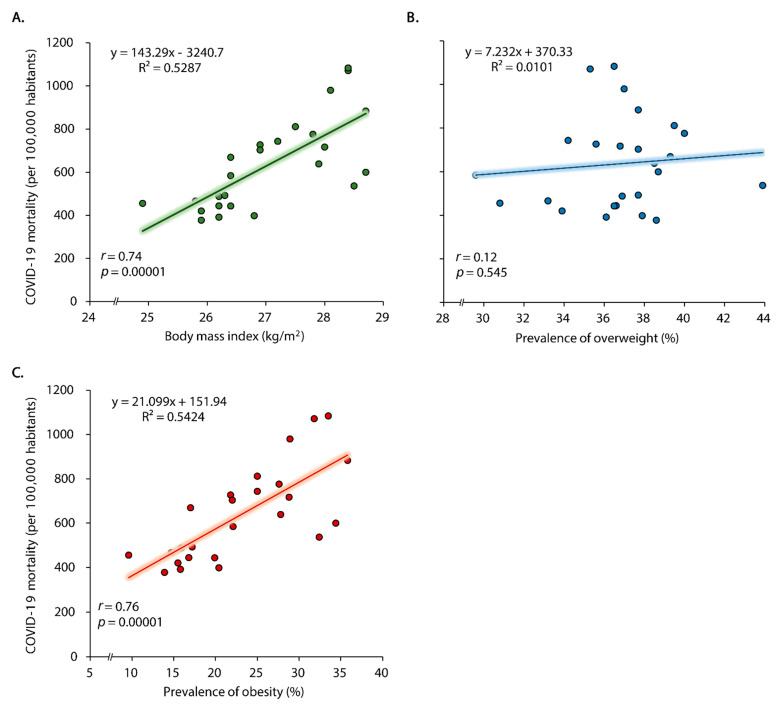
Correlation between body mass index (BMI) average (**A**), prevalence of overweight (**B**) and obesity (**C**) and mortality rates due to COVID-19 in Peru, March 2020 to June 2021.

**Table 1 tropicalmed-06-00182-t001:** BMI average, prevalence of overweight and obesity, and COVID-19 mortality rate in ≥15-year-olds in 25 Peruvian regions, March 2020 to June 2021.

Region	BMI (kg/m^2^) [18]	Prevalence of Overweight (%) [18]	Prevalence of Obesity (%) [18]	COVID-19 Deaths [19]	Population [20]	COVID-19 Mortality Rate (per 100,000 Habitants)	Mean Age (Years)	Mean Monthly Income (PEN)	Gender Balance (Men/Women) [20]	No. of ICU Beds
Amazonas	26.2	36.1	15.8	1135.0	289,802.0	391.6	30.95	1014.0	1.07	19
Ancash	26.9	35.6	21.8	6373.0	876,703.0	726.9	32.36	1230.9	1.01	62
Apurimac	25.8	33.2	14.7	1401.0	300,395.0	466.4	29.99	1123.8	1.05	38
Arequipa	28.0	36.8	28.8	8519.0	1,187,931.0	717.1	32.45	1703.1	0.95	92
Ayacucho	25.9	33.9	15.5	1949.0	464,136.0	419.9	30.04	970.6	1.04	20
Cajamarca	25.9	38.6	13.9	3839.0	1,016,792.0	377.6	30.71	954.4	0.98	61
Callao	28.4	35.3	31.8	9670.0	902,609.0	1071.3	32.36	1579.6	0.94	113
Cusco	26.2	36.6	16.8	4390.0	988,897.0	443.9	30.34	1234.1	1.02	50
Huancavelica	24.9	30.8	9.6	1079.0	236,955.0	455.4	30.28	742.1	1.01	21
Huanuco	26.2	36.9	15.9	2558.0	524,371.0	487.8	30.01	1007.1	1.01	44
Ica	28.4	36.5	33.5	7863.0	725,610.0	1083.6	30.93	1507.5	0.99	84
Junin	26.4	39.3	17.0	6570.0	982,199.0	668.9	31.18	1206.3	0.98	77
La Libertad	27.9	38.5	27.8	9778.0	1,531,668.0	638.4	31.55	1307.5	0.97	96
Lambayeque	27.5	39.5	25.0	8042.0	991,121.0	811.4	32.24	1203.6	0.93	86
Lima	28.1	37.0	28.9	85,748.0	8,750,417.0	979.9	33.05	1885.9	0.91	845
Loreto	26.4	29.6	22.1	3977.0	680,927.0	584.1	28.50	1231.5	1.08	52
Madre de Dios	28.5	43.9	32.4	727.0	135,428.0	536.8	27.48	1665.0	1.37	25
Moquegua	28.7	37.7	35.8	1374.0	155,545.0	883.3	32.85	1801.5	1.17	28
Pasco	26.3	37.7	17.2	961.0	195,114.0	492.5	30.12	1172.0	1.07	31
Piura	27.2	34.2	25.0	11,414.0	1,535,433.0	743.4	30.42	1146.0	1.01	109
Puno	26.8	37.9	20.4	3603.0	904,267.0	398.4	29.72	876.1	0.96	42
San Martin	26.4	36.5	19.9	2833.0	639,533.0	443.0	29.89	1159.2	1.14	49
Tacna	28.7	38.7	34.4	1821.0	303,701.0	599.6	31.91	1392.3	1.04	32
Tumbes	27.8	40.0	27.6	1489.0	191,850.0	776.1	29.91	1264.3	1.20	17
Ucayali	26.9	37.7	22.0	2933.0	416,932.0	703.5	28.00	1174.4	1.13	37

BMI: body mass index (kg/m^2^). ICU: Intensive Care Unit.

**Table 2 tropicalmed-06-00182-t002:** Multiple regression analysis of prevalence of obesity and COVID-19 mortality adjusted.

Variable	Coef.	SE	*Beta*	*t*	*p*-Value
BMI and COVID-19 mortality					
BMI (kg/m^2^)	94.49	51.60	0.479	1.83	0.083
Mean age in the region (years)	−5.58	32.62	−0.038	−0.17	0.866
Mean monthly income (PEN)	0.262	0.23	0.359	1.10	0.285
Gender balance (men/women)	−707.0	482.3	−0.336	−1.47	0.159
No. of ICU beds	0.028	0.251	0.021	0.11	0.911
Obesity and COVID-19 mortality					
Prevalence of obesity (%)	16.76	7.28	0.585	2.3	0.033
Mean age in the region (years)	−2.52	31.2	−0.017	−0.08	0.937
Mean monthly income (PEN)	0.17	0.23	0.240	0.75	0.464
Gender balance (men/women)	−693.81	462.6	−0.330	−1.5	0.15
No. of ICU beds	0.08	0.24	0.068	0.36	0.72

BMI: body mass index (kg/m^2^). ICU: Intensive Care Unit.

## Data Availability

The data presented in this study are publicly available at: Demographic and Health Survey (ENDES) 2020. Noncommunicable and Communicable Diseases. 2020. Available online: https://proyectos.inei.gob.pe/endes/2020/SALUD/ENFERMEDADES_ENDES_2020.pdf (accessed on 14 September 2021); COVID-19 deaths. National System of Deaths (SINADEF). Available online: https://www.datosabiertos.gob.pe/dataset/fallecidos-por-covid-19-ministerio-de-salud-minsa (accessed on 3 June 2021); INEI, Peruvian population: http://censos2017.inei.gob.pe/redatam/ (accessed on 3 June 2021); Daily Report on Form F500.2, app. for centralized management of the availability of Hospitalization and ICU beds at the national level and of all subsystems (Application F500.2): http://portal.susalud.gob.pe/seguimiento-del-registro-de-camas-f500-2/ (accessed on 17 September 2021); Mean monthly income (PEN): https://www.inei.gob.pe/estadisticas/indice-tematico/income/ (accessed on 17 September 2021).

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
