# Peer review of "Association between Obesity and COVID-19 Mortality in Peru: An Ecological Study"

_tropicalmed, 2021, doi:10.3390/tropicalmed6040182_

Round 1
Reviewer 1 Report
The authors submitted an interesting epidemiological report on the association between mortality rates for COVID-19 and obesity rates across 25 Peruvian regions. The results seem to confirm the role of obesity as a risk factor for death during COVID-19, with regions with the highest COVID-19 mortality rates being the same with highest mean BMI levels and highest overweight or obesity prevalence.
Major limitation of this study is the lack if any adjustment for possible confounding factors. Prevalence of overweight and obesity is influenced by age, sex rates, socio-economic status and these same factors could be associated to mortality rates for COVID-19. Differences in the availability of appropriate health care structure could be also worsen the prognosis in poorest regions, were the prevalence of obesity could be possibly higher. I suggest that authors should repeat their analysis by adding some multivariate models taking into account covariates (like mean age in the region, gender balance, mean income, number of ICU beds and similar).
Author Response
Response-to-reviewers: Manuscript tropicalmed-1352292
We thank the Reviewers for their comments and constructive criticism, we believe that the quality of our manuscript has been significantly improved. We have revised our paper in a point-by-point manner. Modifications are in yellow text.
Reviewer #1: Major limitation of this study is the lack if any adjustment for possible confounding factors. Prevalence of overweight and obesity is influenced by age, sex rates, socio-economic status and these same factors could be associated to mortality rates for COVID-19. Differences in the availability of appropriate health care structure could be also worsen the prognosis in poorest regions, were the prevalence of obesity could be possibly higher. I suggest that authors should repeat their analysis by adding some multivariate models taking into account covariates (like mean age in the region, gender balance, mean income, number of ICU beds and similar).
Response: Thank you for your comment. We have included your suggestions. We update the study data, ENDES 2020 includes a sample of 32,197 men and women aged 15 years or more, from 25 Peruvian regions, from January to December 2020.
We explored the association between body mass index (BMI), the prevalence of overweight and obesity, and COVID-19 mortality in 25 Peruvian regions, adjusted by for possible confounding factors. Confounding factors included the mean age in the region (years), mean monthly income (PEN), gender balance and number of ICU beds.

Reviewer 2 Report
Estimated Authors,
The present paper has assessed through an ecological design the prognosis of Sars-CoV-2 in obese individuals.
The overall results confirm an increased risk for a dismal prognosis among obese subjects. This outcome Is nothing new, as It was reported since the earlier studies from Wuhan in the First months of 2020.
However, the present study, through data of good quality and an appropriate study design reaffirm the association between overweight and worse outcome in Sars-CoV-2 patients even in the andine region.
From the point of view of this reviewer, the study does bite require specifico interventions in terms of improved analyses. On the other hand, Authors should include in their discussion the topic represented by the ecological design, and its implicit limits. Such theme has been, in the current versione of the paper, Only marginally addressed.
Author Response
Response-to-reviewers: Manuscript tropicalmed-1352292
We thank the Reviewers for their comments and constructive criticism, we believe that the quality of our manuscript has been significantly improved. We have revised our paper in a point-by-point manner. Modifications are in yellow text.
Reviewer #2: From the point of view of this reviewer, the study does bite require specifico interventions in terms of improved analyses. On the other hand, Authors should include in their discussion the topic represented by the ecological design, and its implicit limits. Such theme has been, in the current versione of the paper, Only marginally addressed.
Comment 1: From the point of view of this reviewer, the study does bite require specifico interventions in terms of improved analyses.
Response 1: Thank you for your comment. We have included your suggestions. We update the study data, ENDES 2020 includes a sample of 32,197 men and women aged 15 years or more, from 25 Peruvian regions, from January to December 2020. We explored the association between body mass index (BMI), the prevalence of overweight and obesity, and COVID-19 mortality in 25 Peruvian regions, adjusted by for possible confounding factors. Confounding factors included the mean age in the region (years), mean monthly income (PEN), gender balance and number of ICU beds.
Comment 2: On the other hand, Authors should include in their discussion the topic represented by the ecological design, and its implicit limits. Such theme has been, in the current versione of the paper, only marginally addressed.
Response 2: Thank you for your comment. We have included your suggestions (See limitations paragraph, Discussion section).
Round 2
Reviewer 1 Report
None